# A complex IRES at the 5'-UTR of a viral mRNA assembles a functional 48S complex via an uAUG intermediate

**Ritam Neupane[1,2†], Vera P Pisareva[3†], Carlos F Rodriguez[4], Andrey V Pisarev[3]\*, Israel S Fernández[2]\***

[1]Department of Biological Sciences, Columbia University, New York, United States; [2]Department of Biochemistry and Molecular Biophysics, Columbia University, New York, United States; [3]Department of Cell Biology, SUNY Downstate Medical Center, Brooklyn, United States; [4]Structural Biology Programme, Centro Nacional de Investigaciones Oncológicas (CNIO), Madrid, Spain

**\*For correspondence:**
andrey.pisarev@downstate.edu (AVP);
isf2106@cumc.columbia.edu (ISFá)

[†]These authors contributed equally to this work

**Competing interests:** The authors declare that no competing interests exist.

**Abstract** Taking control of the cellular apparatus for protein production is a requirement for virus progression. To ensure this control, diverse strategies of cellular mimicry and/or ribosome hijacking have evolved. The initiation stage of translation is especially targeted as it involves multiple steps and the engagement of numerous initiation factors. The use of structured RNA sequences, called Internal Ribosomal Entry Sites (IRES), in viral RNAs is a widespread strategy for the exploitation of eukaryotic initiation. Using a combination of electron cryo-microscopy (cryo-EM) and reconstituted translation initiation assays with native components, we characterized how a novel IRES at the 5'-UTR of a viral RNA assembles a functional initiation complex via an uAUG intermediate. The IRES features a novel extended, multi-domain architecture, that circles the 40S head. The structures and accompanying functional data illustrate the importance of 5'-UTR regions in translation regulation and underline the relevance of the untapped diversity of viral IRESs.

## Introduction

Metagenomic studies of environmental samples have uncovered a great diversity of viruses that have a pervasive presence in the biosphere (*Zhang et al., 2019*; *Zhang et al., 2018*; *Greninger, 2018*). This diversity is especially overwhelming in RNA viruses that infect animal hosts (*Shi et al., 2016*; *Dolja and Koonin, 2018*). As strict cellular parasites, viruses rely on capturing cellular ribosomes to gain access to the host machinery for protein production (*Jan et al., 2016*). In eukaryotes, especially in animals, this machinery is complex and sophisticated, involving large, multi-component protein factors that assist in the operation of eukaryotic ribosomes (*Hashem and Frank, 2018*). Although complex, translation in eukaryotes conserves four main phases that are also found in its prokaryotic counterparts, namely: initiation, elongation, termination and recycling (*Schmeing and Ramakrishnan, 2009*). Initiation is significantly expanded in eukaryotes, with two GTP-regulated steps required for the correct positioning of the first aminoacyl-tRNA responsible for setting up the correct reading frame on the messenger RNA (mRNA) (*Jackson et al., 2010*; *Aitken and Lorsch, 2012*; *Myasnikov et al., 2009*; *Hinnebusch, 2014*).

Eukaryotic initiation starts when the 40S subunit together with the initiation factors eIF1, eIF1A, eIF3, eIF5 and the Ternary Complex (TC) (eIF2–Met-tRNA$_i^{Met}$–GTP) form the 43S Pre-Initiation Complex (43S-PIC), which is competent for mRNA recruitment (*Jackson et al., 2010*). Eukaryotic mRNAs are then docked to the 43S-PIC at their 5' ends, forming the 48S complex (*Hinnebusch, 2017*). Once the AUG codon is detected, a structural transition in the 48S from an open, scanning-competent conformation to a closed, scanning-arrested conformation occurs (*Hussain et al., 2014*). This

conformational change is accompanied by the release of eIF1, eIF2 and GDP, leaving the Met-tRNA$_i$-$^{Met}$ at the P-site of the 40S base paired with the AUG codon (*Aitken and Lorsch, 2012*). A second GTP-regulated step, catalyzed by initiation factor eIF5B, is then required for the recruitment of the large (60S) ribosomal subunit (*Pestova et al., 2000*; *Lee et al., 2002*). A full (80S) ribosome primed with mRNA and Met-tRNA$_i$$^{Met}$ at the P-site then transitions to the elongation phase (*Wang et al., 2019*; *Voorhees and Ramakrishnan, 2013*).

The pathway described above is called the canonical, 5'-end and cap-dependent translation route of initiation (*Hinnebusch, 2014*). The bulk of eukaryotic mRNAs transitions follow this route, but deviations from the canonical route are common, and normally associated with translation under stress conditions (*Starck et al., 2016*; *Shatsky et al., 2010*). Non-canonical initiation is also associated with extended 5' UnTranslated Regions (5'-UTRs) on mRNAs (*Sendoel et al., 2017*; *Young and Wek, 2016*). In complex eukaryotes, 5'-UTRs can be very long and can harbor short Open Reading Frames (ORFs) designated as upstream ORFs (uORFs) (*Young and Wek, 2016*; *Wethmar, 2014*). Well-studied examples of the functional relevance of uORFs at 5'-UTRs can be found in the yeast stress response regulator GCN4 or the mammalian transcription factor ATF4 (*Hinnebusch, 1993*; *Vattem and Wek, 2004*). uAUG codons that are immediately followed by a stop codon (designated as 'start-stop uORFs') are also found in the 5'-UTRs of mammalian mRNAs (*Wethmar, 2014*; *Gunišová et al., 2018*), but little is known about how these 'start-stop uORFs' regulate translation.

Viruses exploit the complexity of eukaryotic initiation to gain access to the host machinery for protein production (*Jaafar and Kieft, 2019*). Strategies such as mimicking the cap structure or transferring caps from cellular mRNAs ('cap-snatching') allow viral mRNAs to hijack host ribosomes, redirecting them towards the production of viral proteins (*Jan et al., 2016*; *Jaafar and Kieft, 2019*). A more prominent viral strategy for ribosome hijacking is the use of structured RNA sequences in viral mRNAs (*Yamamoto et al., 2017*). These sequences are called Internal Ribosomal Entry Sites (IRES), and a tentative classification based on their degree of RNA structure and dependency on canonical initiation factors divided them in four main types (*Filbin and Kieft, 2009*; *Johnson et al., 2017*).

The *Dicistroviridae* family of positive single-stranded RNA ((+)-ssRNA) viruses employs two types of IRESs to express the regulatory versus the structural genes differentially (*Nakashima and Uchiumi, 2009*). The genome architecture of these viruses functionally segregates both kinds of genes in two ORFs (*Figure 1A*; *Hertz and Thompson, 2011*). The first ORF is preceded by an approximately 700-nucleotide 5'-UTR, which harbors an IRES assigned to the type III family (*Gross et al., 2017*). In vitro characterization of the 5'-UTR-IRES of the Cricket Paralysis Virus (CrPV), a prototypical *Dicistrovirus,* narrowed down the region of the 5'-UTR responsible for the IRES activity and established the strict requirement of eIF3 for this IRES to initiate translation. Interestingly, the AUG codon of the CrPV ORF1 is immediately preceded by a 'start-stop uORF' (*Gross et al., 2017*).

We sought to characterize the structure of the 5'-UTR-IRES of the CrPV in its ribosome-bound configuration, to gain insights relating to the ribosome-binding determinants of this peculiar IRES, and to understand how the delivery of Met-tRNA$_i$$^{Met}$ is accomplished. Two high-resolution cryo-EM reconstructions of 40S–5'-UTR-IRES–eIF3 complexes, combined with biochemical analysis, allowed us to characterize how this IRES uses an extended structure with a modular, multi-domain architecture to bind to and manipulate the 40S.

## Results

### The 5'-UTR-IRES of the CrPV requires eIF3 for a stable interaction with the 40S

Previous studies of the IRES located at the 5'-UTR of the CrPV (hereafter referred to as 5'-UTR-IRES) precisely defined the region of the 5'-UTR that is responsible for the IRES activity (residues 357 to 709), as well as its dependency on initiation factor eIF3 for efficient translation initiation (*Gross et al., 2017*). In contrast to the well-characterized type IV family of IRESs found in the InterGenic Region (IGR-IRES) of these viruses, the 5'-UTRs of *Dicistroviruses* seem to harbor divergent sequences, making structural modelling based on sequence conservation difficult (*Kieft, 2009*). In order to address this gap in knowledge, we produced a truncated version of the 5'-UTR region of the genomic RNA of the CrPV that contains the IRES (residues 357 to 728, *Figure 1A*) in order to obtain structural information about its 40S-bound conformation by electron cryo-microscopy (cryo-

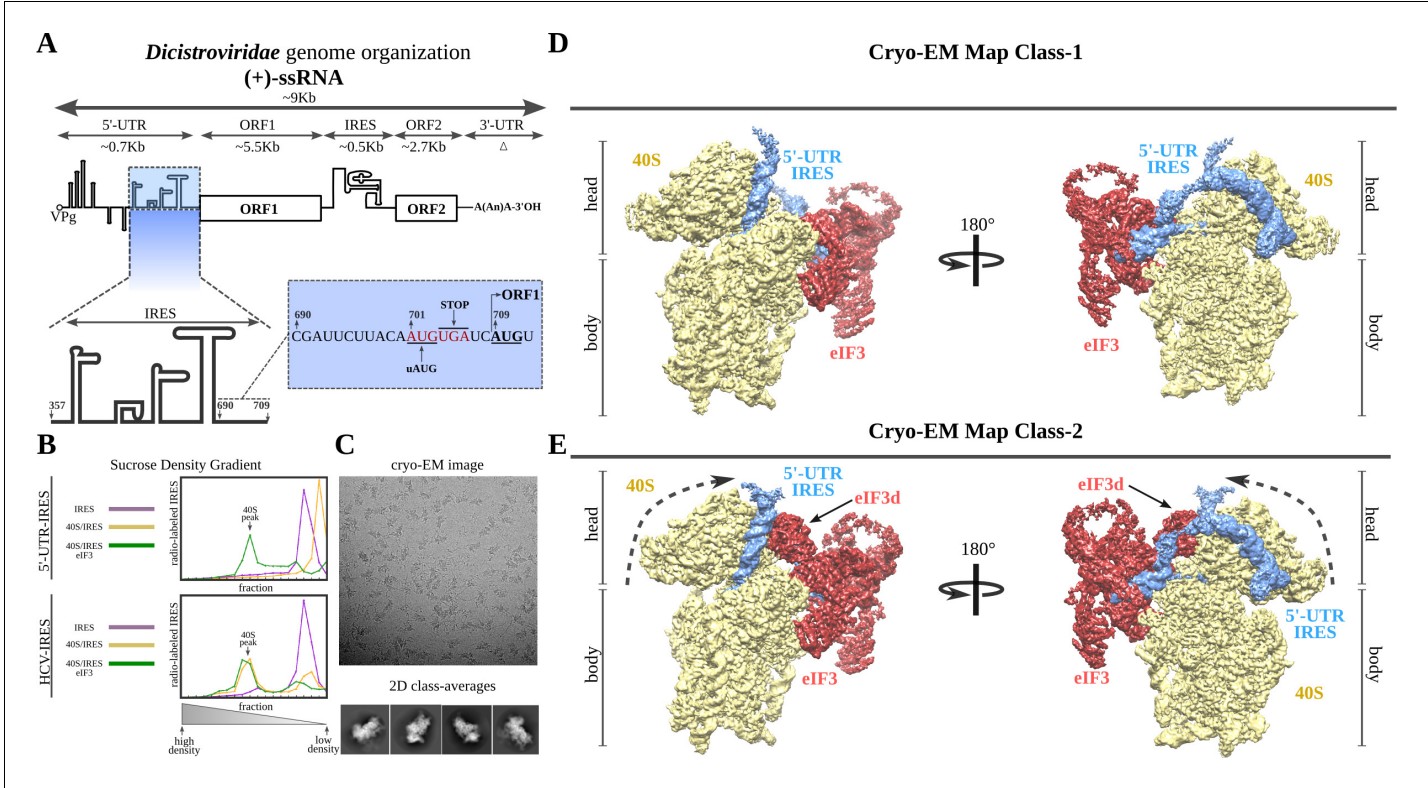

**Figure 1.** *Dicistroviridae* genome organization, in vitro complex formation and cryo-EM maps. (A) Top, schematic representation of the genome organization of *Dicistroviruses*. The approximate genomic lengths of the different components are indicated by the arrows. Bottom, detailed view of the region described as harboring the IRES activity of the 5'-UTR of the CrPV. On the right, the sequence adjacent to the initiation AUG codon of ORF1, located at nucleotide 709 and preceded by a 'start-stop uORF' indicated in red. (B) Sucrose-gradient analysis showed that the 5'-UTR-IRES is dependent on eIF3 in order to form a stable complex with the 40S. 5'-UTR-IRES co-migrates with the 40S only in the presence of eIF3 (top). By contrast, HCV IRES does not require eIF3 for 40S binding (bottom). (C) Representative cryo-EM micrograph of the 40S–5'-UTR-IRES–eIF3 complex. Bottom, representative reference-free 2D class averages used for further image processing. (D, E) After 3D classifications, two classes showing density for 40S (yellow), eIF3 (red) and 5'-UTR-IRES (blue) could be identified in the dataset. Class-1 (D) presents a non-swiveled configuration of the 40S head and the density for eIF3d is absent. Class-2 (E) shows a swiveled configuration of the 40S head (arrows) with eIF3d (indicated) contacting eIF3's core subunits.

The online version of this article includes the following figure supplement(s) for figure 1:

**Figure supplement 1.** Cryo-EM representative images and classification workflow.

**Figure supplement 2.** FSC correlation curves, local resolution and model validation.

EM). We initially tested the in vitro dependency of 5'-UTR-IRES on eIF3 when engaging purified 40S ribosomal subunits in a stable interaction. We assayed the ability of the 5'-UTR-IRES to co-migrate with purified 40S in sucrose density gradients as a test for the presence of a stable complex that is suitable for structural studies (*Figure 1B*). Unexpectedly, the 5'-UTR-IRES does not form a stable complex with the 40S in the absence of eIF3, in contrast to the HCV-IRES, which is able to form stable complexes with the 40S subunit alone and even with full (80S) ribosomes (*Figure 1B*; *Yokoyama et al., 2019*). Sucrose density gradients were manually fractionated from the bottom, where density is heavier. This caused small variations in the position of the 5'-UTR-IRES at the top of the gradients among different experiments. We do not believe such shifts have functional implications.

In the presence of eIF3, however, the 5'-UTR-IRES co-migrates with purified 40S subunits (*Figure 1B*). This complex revealed clear particles in cryo-EM images, rendering detailed two-dimensional class averages in which density for eIF3 could be identified, albeit at lower threshold (*Figure 1C* and *Figure 1—figure supplement 1*). The 40S–5'-UTR-IRES–eIF3 complex exhibited a delicate behavior under cryo-EM conditions, with a strong tendency to disassemble in thin ice. Extensive screening for suitable ice areas was essential to obtain particles of the fully assembled

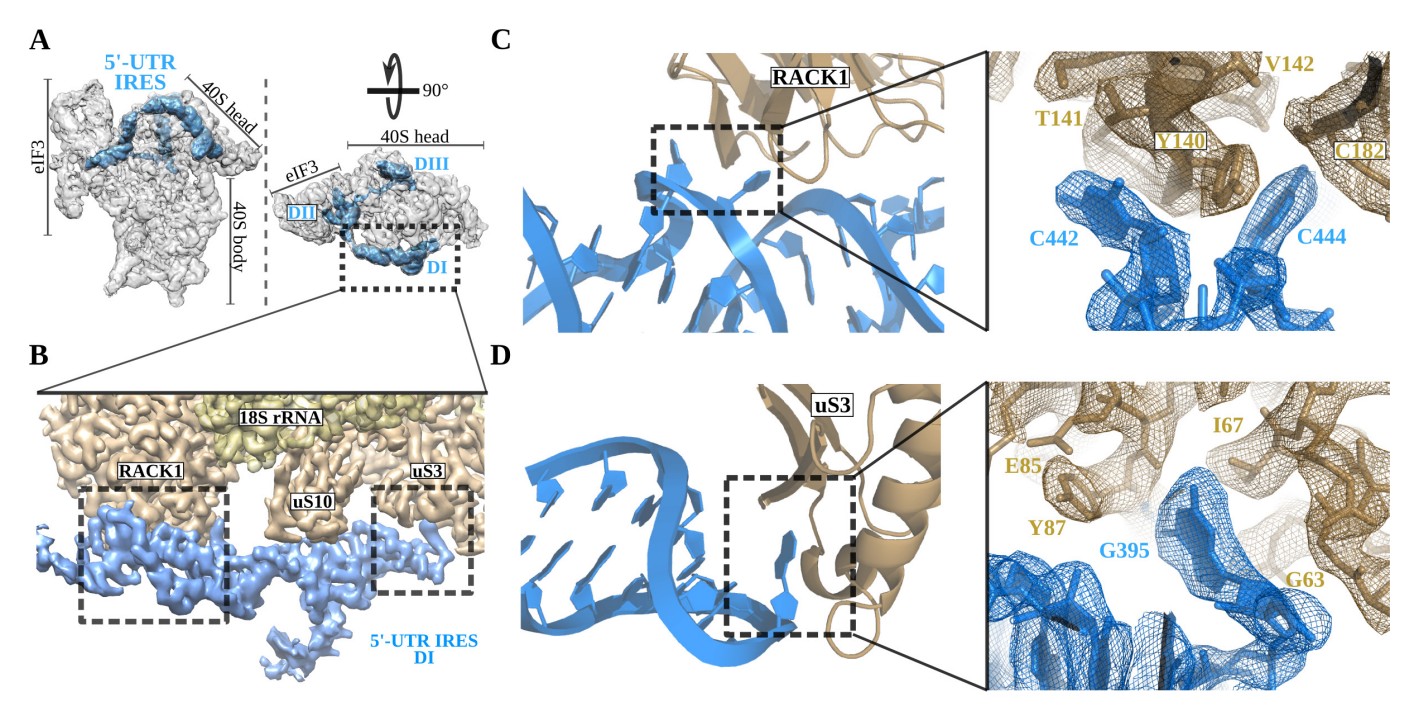

**Figure 2.** The 5'-UTR-IRES domain I engages ribosomal proteins RACK1 and uS3.  (**A**) Overview of the 40S–5'-UTR-IRES–eIF3 map, with 40S and eIF3 depicted gray and 5'-UTR-IRES in blue. (**B**) Detailed view of the cryo-EM density of the 40S–5'-UTR-IRES–eIF3 map centered around 5'-UTR-IRES domain I (DI). Ribosomal proteins are colored brown, 18S rRNA yellow and 5'-UTR-IRES blue. Contacts between 5'-UTR-IRES domain I and ribosomal proteins RACK1 (**C**) and uS3 (**D**) could be defined thanks to well-resolved local cryo-EM densities.

The online version of this article includes the following figure supplement(s) for figure 2:

**Figure supplement 1.** Structurally derived secondary structure diagram for the 5'-UTR-IRES.

complex (*Figure 1C* and *Figure 1—figure supplement 1*). The sample also exhibited a high degree of heterogeneity, which could be resolved by image processing in Relion (*Scheres, 2012*; *Scheres, 2016*; *Figure 1D,E* and *Figure 1—figure supplement 2*).

Two main classes of particles containing density for 5'-UTR-IRES, 40S and eIF3 were found in the dataset (*Figure 1D and E*). Both classes contain density for the 40S, the IRES and the core subunits of eIF3 (a/c/e/k/l/f/m), and class-2 also presents density for eIF3 subunit d (*Figure 1E*, eIF3d). Class-2 exhibits a 40S head in a swiveled configuration. Owing to this swiveled configuration of the 40S head, eIF3d establishes interactions with eIF3a, a core subunit of eIF3 (see below).

Robust density ascribable to the 5'-UTR-IRES could be found in both classes of complex (*Figure 1D and E*, blue). The ribosome-bound conformation of 5'-UTR-IRES shows an extended configuration, almost circling the 40S head (*Figure 2A* and *Figure 2—figure supplement 1*). Three domains connected by flexible linkers could be defined: an elongated domain I (DI) at the back of the 40S head contacting ribosomal proteins uS3 and RACK1 (*Figure 2*), a second domain (DII) formed by a dual hairpin at the back of the 40S body interacting with eIF3 (*Figure 3*), and a third, large helical domain (DIII) placed at the periphery of the 40S E-site, contacting ribosomal proteins uS7 and uS11 (*Figure 4*).

## Domain I of the 5'-UTR-IRES contacts the ribosomal proteins RACK1 and uS3

The 5' proximal segment of the 5'-UTR-IRES (residues 357 to 486) forms domain I, which is characterized by an elongated T-shaped structure anchored to the back of the 40S head (*Figure 2A and B*). A long helical segment in this domain 'wraps' around the apical part of ribosomal protein RACK1. Two bases of this helical segment of domain I, C442 and C444, are extruded from the body of the double helix to establish hydrophobic interactions with tyrosine residue 140 of

RACK1 (*Figure 2C*). These interactions bend the main helical segment of the 5'-UTR-IRES DI, directing the tip of this domain towards ribosomal protein uS3 (*Figure 2D*). Guanine residue 395 is inserted deep into a hydrophobic pocket of ribosomal protein uS3, establishing contacts with main-chain atoms of this ribosomal protein. In this location, 5'-UTR-IRES DI is found adjacent to the mRNA entry channel of the 40S, overlapping with the space previously described as being occupied by the helicase DHX29 involved in canonical initiation (*Figure 2D*; *Hashem et al., 2013a*).

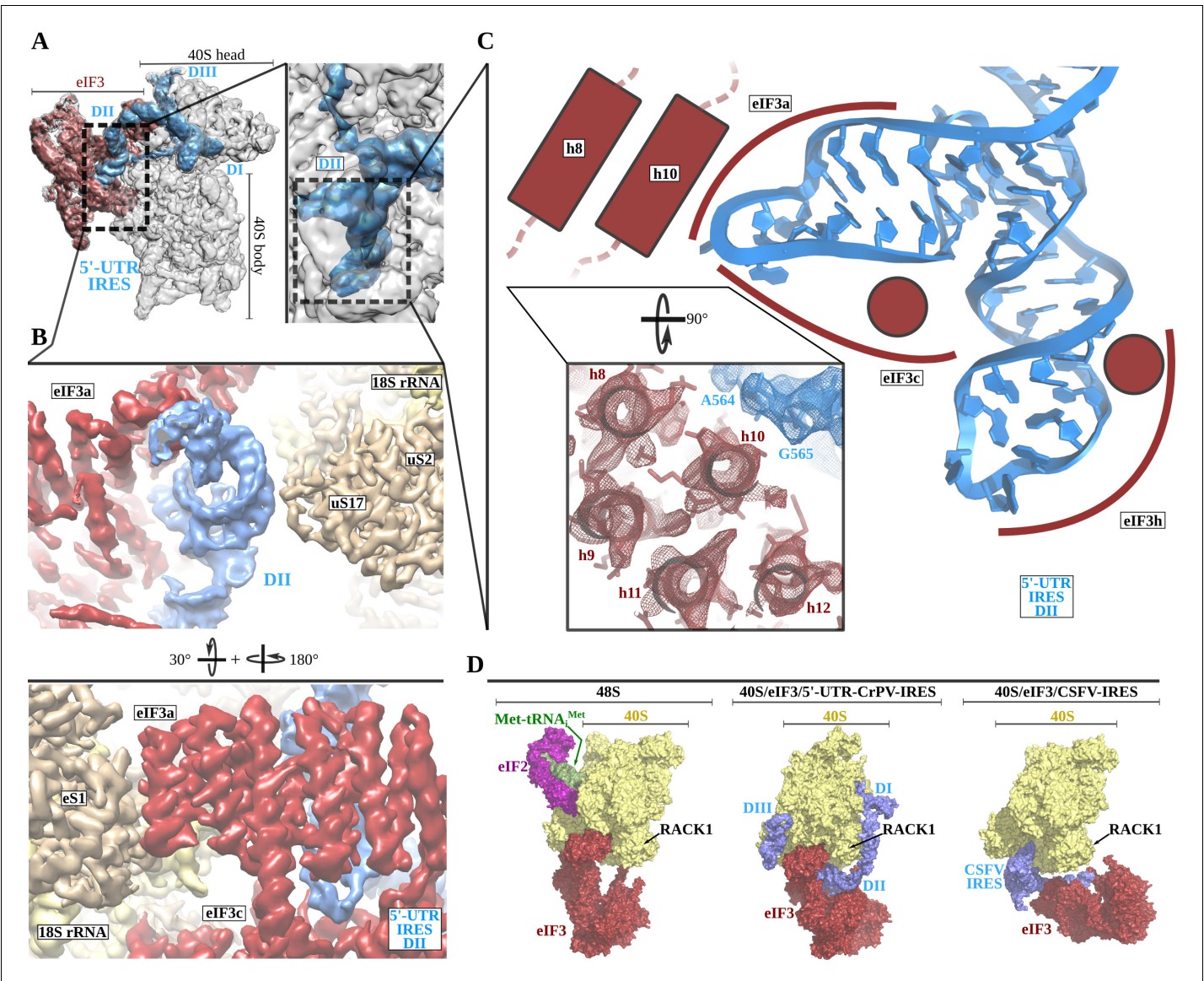

**Figure 3.** The 5'-UTR-IRES domain II is formed by a dual hairpin that mediates eIF3 recruitment. (**A**) Overview of the 40S–5'-UTR-IRES–eIF3 cryo-EM map with 40S colored gray, eIF3 red and 5'-UTR-IRES blue. On the right, a zoomed view centers around 5'-UTR-IRES domain II. (**B**) Detailed view of the cryo-EM map for the region occupied by 5'-UTR-IRES domain II, with 40S components colored gold, eIF3 red and 5'-UTR-IRES blue. Domain II is sandwiched between ribosomal protein uS17 (located at the back of the 40S body) and eIF3 core subunits a and c. (**C**) 5'-UTR-IRES domain II is formed by a dual hairpin that establishes interactions with α-helices 8 and 10 from eIF3a. These contacts are mediated mainly by basic residues of eIF3 and the phosphate backbone of the IRES. (**D**) superposition of the 40S–5'-UTR-IRES–eIF3 complex with the canonical 48S complex (left, PDB ID 6FEC) and with the CSFV-IRES–40S complex (right, PDB ID 4c4q). The 5'-UTR-IRES binds to the 40S with a conformation that is compatible with the canonical position described for eIF3 in the 48S complex.

## 5'-UTR-IRES binding to the 40S is compatible with a canonical configuration of eIF3

The second domain of 5'-UTR-IRES (DII) is connected to domain I by a flexible linker that is poorly defined in our maps as it is exposed to the solvent. This second domain of the 5'-UTR-IRES is formed by a dual hairpin and is wedged between the back of the body of the 40S and eIF3 subunits a and c (*Figure 3A and B*). Ribosomal protein uS17 peripherally contacts this domain, establishing interactions with the phosphate backbone of the IRES (*Figure 3B*). A network of interactions involving eIF3 subunits a, c and h anchors DII to this position (*Figure 3C*). These interactions are also established through contacts between positively charged residues on eIF3 and the phosphate backbone of the IRES. No contacts with specific IRES bases could be observed.

Currently, medium-resolution cryo-EM reconstructions for 40S complexes containing eIF3 and the rest of the components of the canonical 48S complex are available (*Eliseev et al., 2018*) (PDB ID 6FEC), as are such reconstructions for the 40S in complex with eIF3 and the CSFV-IRES (*Hashem et al., 2013b*) (PDB ID 4c4q). Comparisons of these structures with our complex reveal a positioning of eIF3 relative to the 40S that is very similar to the canonical 48S complex and different from the position adopted by eIF3 in the CSFV-IRES–40S complex (*Figure 3D*). In the 48S canonical configuration, eIF3 contacts the 40S through helix 1 of eIF3a and helix 22 of eIF3c, as well as through eIF3d, which is isolated in its 40S interaction, away from the core subunits of eIF3 (*Figure 3D*, left). The CSFV-IRES engages the 40S, displacing eIF3 from its position in the canonical 48S (*Figure 3D*, right). In addition, in the canonical 48S complex, eIF3 interacts with the 40S peripherally, allowing the formation of cavities between eIF3 and the back of the 40S. These cavities are exploited by the 5'-UTR-IRES, which inserts its domain II into one of them, adopting a configuration that is compatible with the binding of eIF3 to the 40S in the canonical 48S complex (*Figure 3D*, middle). No major rearrangement of eIF3 (compared to its position in the canonical 48S complex) is required for the binding of the 5'-UTR-IRES, so there could be an advantage in hijacking preformed cellular 48S complexes that are ready to transit the cap-dependent route of initiation.

## Non-canonical base pairing in the 5'-UTR-IRES DIII places the uAUG codon near the P-site

Threading through the 40S channel formed by ribosomal proteins uS7 and uS11, a flexible single stranded linker connects DII with DIII (*Figure 4A*). DIII forms a prominent, helical mass in the surroundings of the E-site of the small subunit at the inter-subunit face of the 40S. The helical segment is very well defined in our maps because it is stabilized by numerous contacts with ribosomal proteins uS7 and uS11 and with 18S ribosomal RNA (rRNA) bases (*Figure 4* and *Figure 2—figure supplement 1*). However, the distal part of this domain forms two short stem loops that, given their flexibility, could only be modelled at low resolution.

Inspection of the cryo-EM density reveled a distortion in the canonical double helix of the main segment of this domain as it approaches the E-site. The quality of the maps in this area allowed de novo modelling of these residues, revealing a set of non-canonical interactions between the RNA bases (*Figure 4A and B*). In-plane triple-base interactions involving sugar and the Hoogsteen edges of the bases, as well as purine–purine Hoogsteen base pairs, could be found in this stretch of residues of the helical segment of DIII (*Figure 4B*; *Leontis and Westhof, 1998*). Overall, these non-canonical base pairs induce a distortion at the base of DIII that helps to position the single-stranded segment of the 5'-UTR-IRES harboring the uAUG codon at position 701 in the mRNA-binding channel of the 40S (*Figure 4C*, middle). The 5'-UTR-IRES accesses the 40 S P-site through the E-site, blocking a concurrent recruitment of the TC (eIF2–Met-tRNA$_i^{Met}$–GTP, *Figure 4C*). Interestingly, a similar strategy is followed by the HCV-IRES. A superposition of the structure of the HCV-IRES in complex with the 40S (*Yamamoto et al., 2015*; *Quade et al., 2015*) (PDB ID 5A2Q) with our structure reveals a very similar positioning of the domain II of HCV-IRES, accessing the P-site through the E-site to position the AUG codon in the surroundings of the P-site (*Figure 4C*, bottom). Even though both IRESs differ markedly in their interaction with the back of the 40S and eIF3, both converge to similar structural solutions for the placement of the AUG initial codon close to the 40 S P-site.

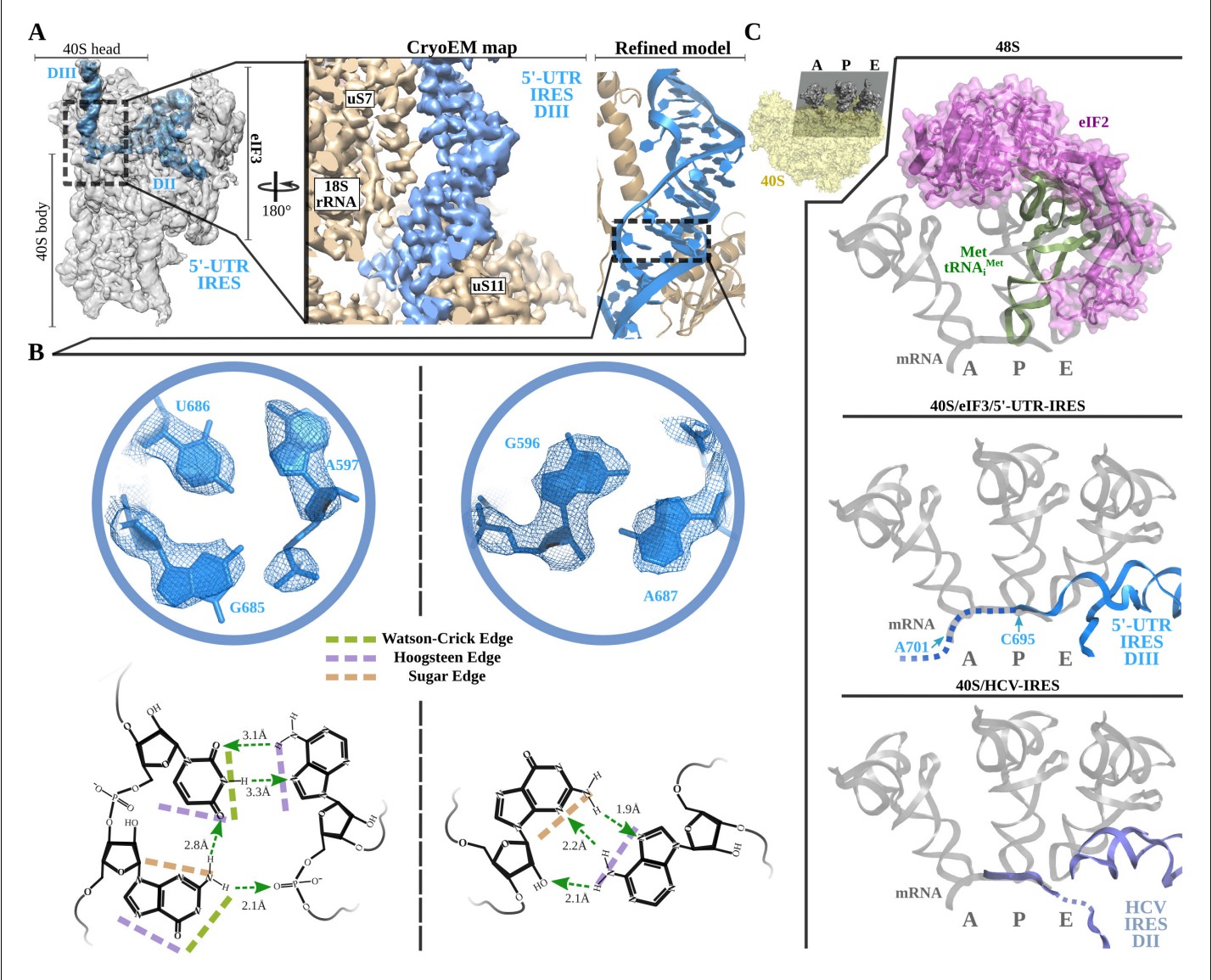

**Figure 4.** Non-canonical base pairing in 5′-UTR-IRES domain III assists on P-site access. A) Overview of the 40S–5′-UTR-IRES–eIF3 cryo-EM map with 40S and eIF3 colored gray and 5′-UTR-IRES blue. On the right, a detailed view of the E-site, where 5′-UTR-IRES domain III is placed, shows the cryo-EM map with the 5′-UTR-IRES colored blue and 40S components brown. On the far right, the final refined model is colored following the same color scheme. Ribosomal proteins uS7 and uS11 as well as several 18S rRNA bases contact 5′-UTR-IRES domain III. (B) Non-canonical base pairs found in 5′-UTR-IRES domain III induce a distortion of the double helix near the E-site. At the top, two examples are shown with the refined model inserted in the experimental cryo-EM density. The corresponding chemical diagrams below show the base edges and the hydrogen bonds involved in interactions. (C) Superpositions of the canonical 48S complex (PDB ID 6FEC, top) with the HCV-IRES/40S complex (PDB ID 5A2Q, bottom) and the 40S/5′-UTR-IRES/eIF3 (middle) models, focused on the tRNA A, P and E binding sites. 5′-UTR-IRES domain III and HCV-IRES occupies a space on the E-site that overlaps with the position described for eIF2 in the canonical 48S complex. In the middle, the last residue of 5′-UTR-IRES is indicated (C695), as is the putative path along the mRNA binding channel followed by the IRES (dashed line).

## Swiveling of the 40S head locks the 5′-UTR-IRES, inducing a compact conformation of eIF3

Initial processing of the cryo-EM data revealed flexibility of the 40S head. Masked classification and refinement in Relion3 (*Zivanov et al., 2018*) revealed two major populations of particles, which differ in the degrees of 40S head swiveling (*Figure 1*). The 40S head is attached to the body by a single RNA helix, making this component of the ribosome extremely flexible (*Johnson et al., 2017*). Intrinsic and independent movements of the 40S head are instrumental in tRNA translocation

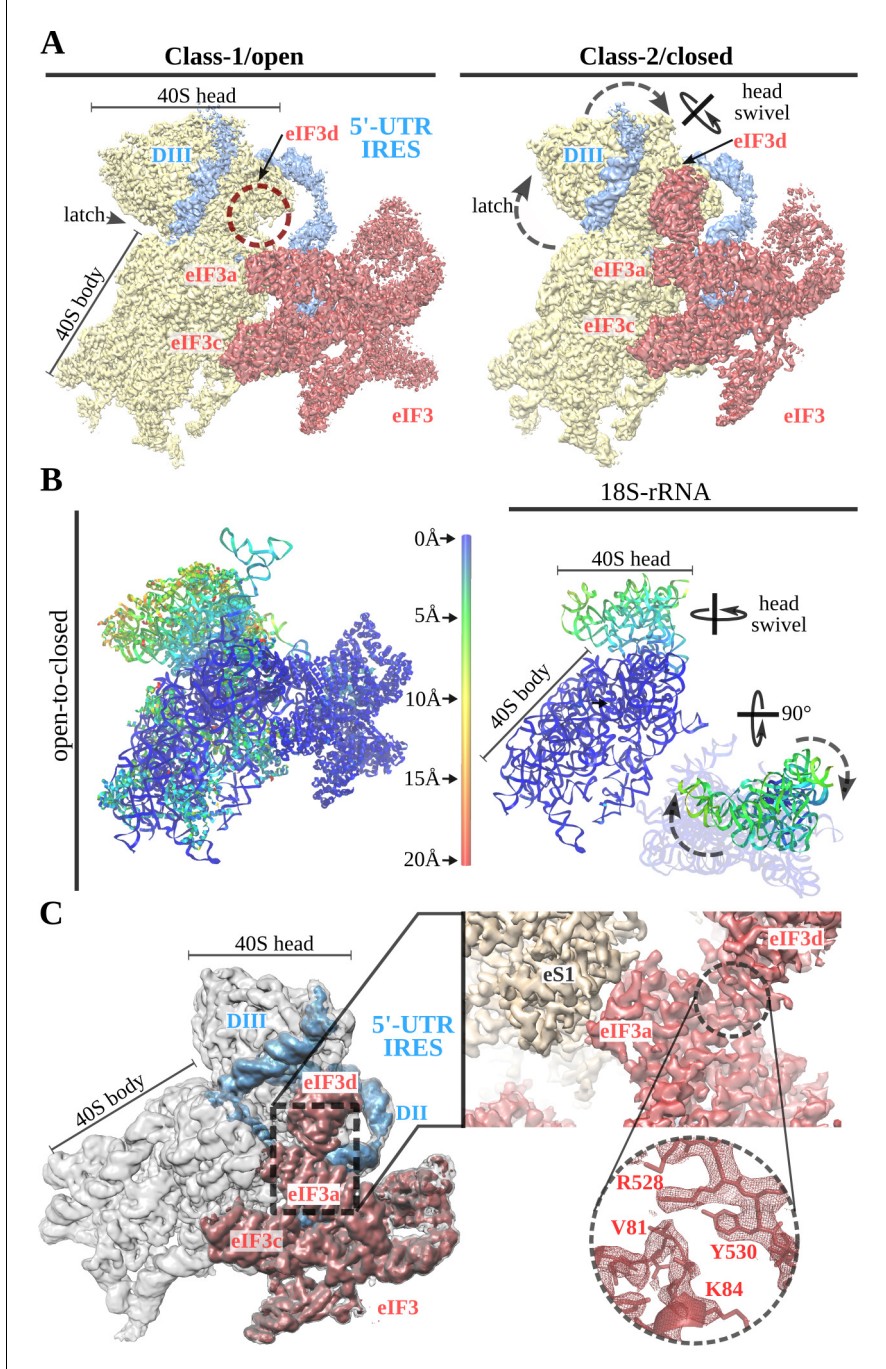

**Figure 5.** A 40S head swiveling movement 'locks' 5'-UTR-IRES on the 40S, inducing a compact eIF3 configuration. (**A**) Cryo-EM maps obtained for the two classes present in the 40S–5'-UTR-IRES–eIF3 dataset with 40S colored gold, eIF3 red and 5'-UTR-IRES blue. The positions of the latch, eIF3d and the swiveling rotation axis are indicated. A 40S head swiveling movement in class-2 brings eIF3d closer to the core subunit of eIF3 (i. e. eIF3a), establishing interactions that stabilize its conformation. (**B**) Left, ribbon diagram colored according to pairwise <u>r</u>oot <u>m</u>ean <u>s</u>quare <u>d</u>eviation (r.m.s.d.) displacements for the open-to-closed transition, with displacements scale at the center. On the right, a simplified diagram shows only the 18S rRNA colored according to the used on the left. Two orthogonal views are shown, in which it can be appreciated that the main displacement is localized at the 40S head. (**C**) Overview of the closed class with 40S colored gray, eIF3 red and 5'-UTR-IRES blue. Inset, detail of the experimental density obtained for the eIF3a–eIF3d interface for this class. Clear information on side chains was present in the maps, allowing proper model building and refinement.

and also in canonical initiation (*Ratje et al., 2010*; *Flis et al., 2018*). The 5'-UTR-IRES seems to exploit this intrinsic dynamic to bind to the 40S and then to 'lock' the IRES in a specific conformation that commits the complex towards viral translation (*Figure 5*). In class-1 (open conformation), the head of the 40S shows an almost canonical configuration with very little swiveling and no tilt. In this conformation, the latch of the 40S (an early defined contact between the head and the body of the 40S [*Frank et al., 1995*]) is closed. At the other side of the 40S head, access to the channel formed by ribosomal proteins uS7 and uS11 is exposed and eIF3d density is not well defined, probably because of a high degree of flexibility or low occupancy (*Figure 5A*, left). In class-2 (closed conformation), the head of the 40S exhibits a medium-range degree of swiveling when compared to the widest displacement reported (*Ratje et al., 2010*).

In the open and closed classes, the positions of the 5'-UTR-IRES relative to the 40S head are very similar (*Figure 5A*, right and B). In the swiveled conformation, the latch is open, and the channel formed by ribosomal proteins uS7 and uS11 is plugged by eIF3d, which in this class presents robust density (*Figure 5C*). The main subunits of eIF3 (a/c/e/k/l/f/m) show a similar conformation in both classes, having a similar orientation with respect to the 40S body (*Figure 5B*). In the swiveled configuration (class-2), the 40S head brings eIF3d close to eIF3a, one of the core subunits of eIF3 (*Figure 5C*). Well-defined density in this area could be observed for the eIF3a–eIF3d interface (*Figure 5C*, right). This compact state of eIF3 represent a hitherto unknown conformation (*Lee et al., 2016*).

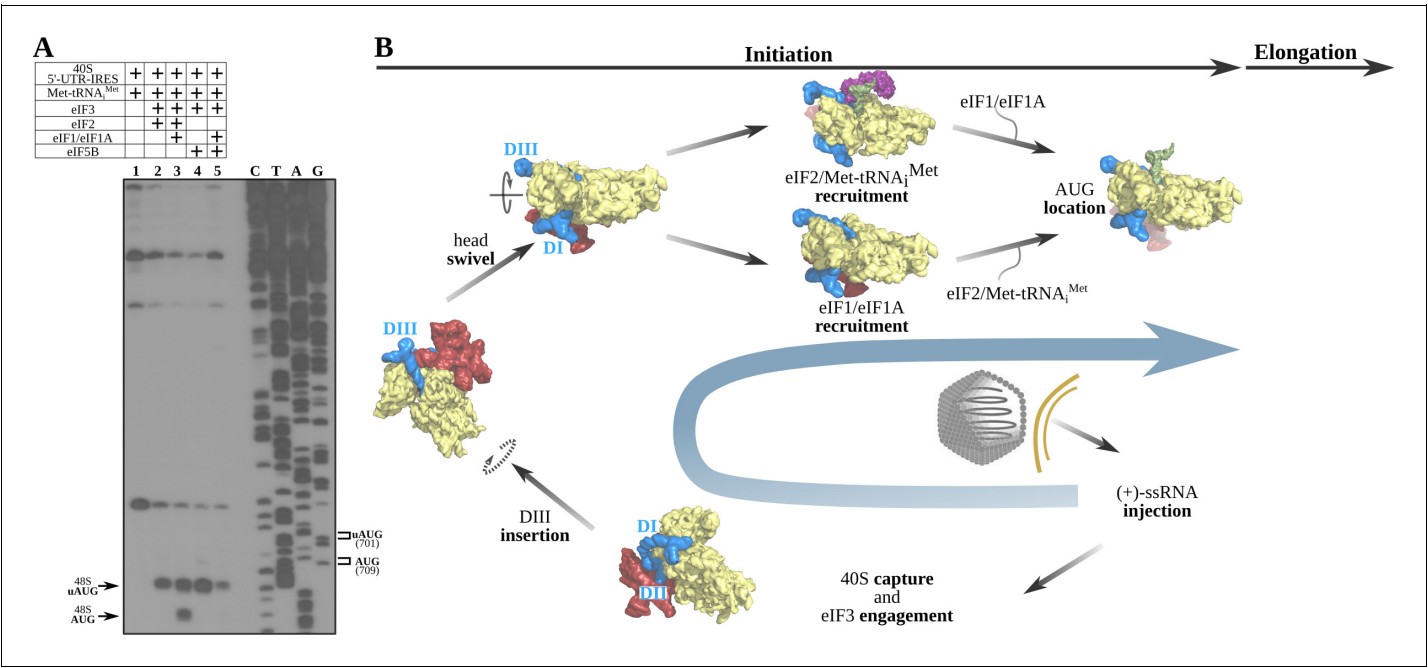

**Figure 6.** The 5'-UTR-IRES requires TC, eIF1 or eIF1A to assemble a functional initiation complex via an uAUG intermediate. (**A**) Toe-print analysis of 48S initiation complexes assembled on 5'-UTR-IRES in an in vitro reconstituted system. eIF2 delivers Met-tRNA$_i^{Met}$ to the uAUG (lane 2) and requires the presence of eIF1 or eIF1A to transition to the *bona fide* AUG codon of ORF1 (lane 3). Under some conditions, eIF5B can substitute for eIF2 in Met-tRNA$_i^{Met}$ delivery (*Terenin et al., 2008*). In the absence of eIF1 or eIF1A, a robust toe-print signal is detected in the presence of eIF5B (lane 4), however, Met-tRNA$_i^{Met}$ is delivered to the uAUG, and thus eIF5B is unable to find the annotated AUG even in the presence of eIF1 or eIF1A (lane 5). (**B**) A model for 5'-UTR-IRES-mediated translation initiation. From the bottom right: injection of the genomic (+)-ssRNA of the CrPV into the cytosol allows the 5'-UTR-IRES to capture free 40S and eIF3, which is recruited to the complex with an initial canonical conformation of the 40S head without tilt or swivel. Insertion of 5'-UTR-IRES DIII in the vicinity of the E-site and a swiveling movement of the 40S head induces a 'locked' conformation of the complex, with the uAUG at 701 in the vicinity of the P-site. Delivery of Met-tRNA$_i^{Met}$ and location of ORF1 AUG is achieved by the concerted action of eIF2, eIF1 and eIF1A. Large subunit recruitment mediated by eIF5B will allow transitioning into elongation.

The online version of this article includes the following figure supplement(s) for figure 6:

**Figure supplement 1.** Control toe-print experiments.

## TC delivers Met-tRNAᵢ$^{Met}$ to uAUG at position 701, and initiation factors eIF1 and eIF1A assist in AUG location

Our structures of the 40S–5'-UTR-IRES–eIF3 complex revealed a positioning of the DIII of the IRES that overlaps with the position that the TC populates at the E-site in canonical initiation (*Figure 4C*; *Hussain et al., 2014*; *Eliseev et al., 2018*). In addition, in our maps, we could only confidently identify density for the single-stranded segment of RNA of the IRES placed close to the P-site until residue 695, whereas the canonical AUG of ORF1 is found at nucleotide 709. These facts prompted us to wonder how the delivery of Met-tRNAᵢ$^{Met}$ to the AUG is accomplished. Making use of an in vitro reconstituted mammalian initiation assay with native components and toe-printing analysis (*Kolupaeva et al., 2007*), we analyzed the different steps followed by the 5'-UTR-IRES in order to place Met-tRNAᵢ$^{Met}$ based paired with the AUG codon at the P-site (*Figure 6A*). Translation initiation factors in mammals and insects are highly homologous. In particular, eIF2-alpha shares 57% identity and 74% similarity between human and *Drosophila*, eIF2-beta 74% identity and 83% similarity, eIF2-gamma 82% identity and 88% similarity, eIF5B 71% identity and 85% similarity, eIF3a 46% identity and 63% similarity, and eIF3c 51% identity and 66% similarity. This high level of homology justifies the utilization of mammalian initiation factors for CrPV analysis, as has been done before for the CrPV IGR-IRES.

Toe-print assay permits identification of the location of functional ribosomal complexes assembled on mRNAs by reverse transcription of a primer annealed to the mRNA. The length of the resulting extended DNA fragment provides information about the position of the ribosome on the mRNA. Due to its large size, the paused ribosome protects a segment of the mRNA, precluding further primer extension and generating toe-print signals approximately 15–17 nucleotides downstream of the P-site of 40S. Cognate aminoacyl or peptidyl-tRNAs in the P-site or post-termination complexes with eRF1 in the A-site yield robust toe-print signals (*Skabkin et al., 2013*). The 40S–5'-UTR-IRES–eIF3 complex produces signal that is ascribable to the secondary structure of the 5'-UTR-IRES (*Figure 6—figure supplement 1*, lanes 1–3), indicating no measurable pausing of the ribosome on the mRNA around any of the AUG codons. In isolation, the TC (eIF2–Met-tRNAᵢ$^{Met}$–GTP) is able to load Met-tRNAᵢ$^{Met}$ onto the P-site of the 40S–5'-UTR-IRES–eIF3 complex, producing a robust toe-print (*Figure 6A*, lane 2, label 48S–uAUG) 15–17 nucleotides away from the uAUG located at 701. A similar uAUG delivery of Met-tRNAᵢ$^{Met}$ can be accomplished by eIF5B which, under stress conditions, has been described as substituting for eIF2 in Met-tRNAᵢ$^{Met}$ delivery (*Terenin et al., 2008*; *Pestova et al., 2008*; *Yamamoto et al., 2014*; *Kenner et al., 2019*), with eukaryotic initiation then following a 'bacterial-like' mode (*Figure 6A*, lane 4). Transitioning to the correct AUG could only be detected in the presence of eIF1and eIF1A, and only when the TC was present, and not for eIF5B (*Figure 6A*, lanes 3 and 5). Notably, the presence of eIF1and eIF1A seems to be detrimental to uAUG Met-tRNAᵢ$^{Met}$ loading by eIF5B, as their presence significantly reduces the toe-print signal that can be observed for eIF5B in isolation. However, no concomitant increase in toe-print signal for the canonical AUG could be observed for the eIF1/eIF1A/eIF5B reaction.

Only eIF2 as part of the TC and assisted by eIF1 and eIF1A can properly locate the *bona fide* AUG of ORF1. The role of the uAUG located at nucleotide 701 is not clear, but the fact that eIF5B can deliver Met-tRNAᵢ$^{Met}$ only to this codon points towards an important role for this uAUG in initiation when eIF2 is unavailable.

## Discussion

Ribosome-profiling datasets have revealed the presence of translating ribosomes paused on 5'-UTRs, implying a decisive role of these sequences in regulating translation, especially under stress conditions (*Sendoel et al., 2017*; *Archer et al., 2016*; *Andreev et al., 2015*; *Ingolia et al., 2009*; *Brar and Weissman, 2015*; *Resch et al., 2009*).

The 5'-UTR of the (+)-ssRNA of the *Dicistovirus* CrPV harbors an IRES that is able to direct initiation towards ORF1 in the early phase of infection (*Hertz and Thompson, 2011*; *Garrey et al., 2010*). Expression of ORF1 is instrumental for virus replication because the RNA-dependent RNA polymerase (RdRp), and the protease responsible for the proteolytic digestion of the polyprotein containing the structural proteins, are encoded in ORF1 (*Jan et al., 2003*).

The 5'-UTR-IRES features a novel multi-domain, extended architecture that encircles three quarters of the 40S head, exploiting binding sites not previously described for any IRESs (*Figures 2*,

*3* and *4*). Ribosomal proteins uS3 and RACK1 are used by the IRES to anchor its DI to the back of the 40S head (*Figure 2*). The structure thus rationalizes previous data showing a preeminent role of RACK1 in CrPV and related viruses that infect *Drosophila* (*Majzoub et al., 2014*). The interaction of DI with RACK1 is also instrumental in positioning DII at the back of the 40S body, sandwiched in between ribosomal protein uS17 and eIF3 (*Figure 3*). Interestingly and in contrast with the HCV-IRES, the conformation observed for eIF3 in the complex with 5'-UTR-IRES is very similar to that observed for eIF3 in the 48S complex, with the IRES 'filling up' cavities that are present between the 40S and eIF3 in this canonical complex (*Eliseev et al., 2018*). The HCV-IRES and related IRESs, such as the CSFV-IRES, displace eIF3 from its canonical location using a very different mechanism for IRES docking to the 40S (*Hashem et al., 2013b*).

In order to place the AUG of ORF1 in the surroundings of the P-site, the 5'-UTR-IRES accesses the P-site through the E-site, in a manner similar to that of the HCV-IRES (*Figure 4C*; *Yamamoto et al., 2015*). In this aspect, the 5'-UTR-IRES recapitulates binding strategies that are known for other IRESs, such as the IGR-CrPV-IRES that also makes use of ribosomal protein uS7 for its binding to the ribosome or the HCV-IRES that places its domains II and IV in the surroundings of the P-site, sliding the elongated DII from the back of the 40S to the P-site through the E-site (*Pisareva et al., 2018*).

The placement of the AUG of ORF1 in the surroundings of the P-site seems to be exerted by a mechanism involving the intrinsic dynamics of the 40S head (*Johnson et al., 2017*; *Figure 5*). The 5'-UTR-IRES exploits the characteristic swiveling movement of the 40S head to bind and progress towards a conformation that 'locks' the IRES onto the 40S, and at the same time, induces a compact conformation of eIF3 that has subunit eIF3d in close contact with the core subunits of eIF3 (*Lee et al., 2016*). These dynamics are probably instrumental for the ability of the 5'-UTR-IRES–40S complex to localize the annotated AUG, in a genomic context where a uAUG-stop is physically close. The capacity of the 40S to scan an mRNA bidirectionally upon termination on a stop codon has been previously reported (*Skabkin et al., 2013*). It is thus plausible that the peculiar genetic configuration of the CrPV around the annotated AUG of ORF1 (*Figure 1A*) evolved to leverage these re-initiation mechanisms already present in the translation of cellular messengers. However, these considerations are highly speculative, as the particular role that the uAUG exerts in Met-tRNA-$_i^{Met}$ recruitment, or more generally its involvement in initiation of viral messengers, remains enigmatic. A comprehensive understanding of the role of uAUG and the start-stop configuration will demand further studies, ideally in vivo.

We propose the following model for how the 5'-UTR-IRES of the CrPV operates: immediately after the (+)-ssRNA genomic molecule of the CrPV is injected into the cytoplasm of the host cell, the IRES harbored at the 5'-UTR captures 40S subunits (*Figure 6B*, bottom). Recruitment of eIF3 is mediated by DII, allowing the sliding of the flexible linker connecting DII and DIII between the head and the platform of the 40S to place DIII in the surroundings of the E-site (*Figure 6B*, bottom and left). A swivel movement of the 40S head closes the channel between the head and the platform of the 40S, effectively 'locking' the 5'-UTR-IRES into the 40S and inducing a compact conformation of eIF3 with the eIF3d subunit in interacting distance with eIF3's core subunit a (*Figure 6B*, left top). With this configuration, eIF2 as part of the TC can deliver Met-tRNA$_i^{Met}$ to the uAUG located at nucleotide 701, and further assistance by initiation factors eIF1 and eIF1A allows for a downstream location of the AUG codon of ORF1 at nucleotide 709. Large subunit recruitment grants transitioning towards elongation, committing the ribosome to the production of viral proteins (*Figure 6B*, right top).

In summary, we have structurally characterized the 5'-UTR-IRES of the CrPV in its ribosome-bound state and have characterized the delivery of Met-tRNA$_i^{Met}$ by eIF2 and eIF5B. Given the rich diversity of viral sequences in the animal virome, new IRESs exploiting different aspects of animal translation will probably be discovered.

## Materials and methods

### 5'-UTR-IRES and HCV IRES production

For cryo-EM analysis, a transcription vector for 5'-UTR-IRES (nucleotides 357–728) was constructed by inserting a T7 promoter sequence upstream of the 5'-UTR-IRES sequence followed by a BamHI restriction site, using pUC19 as a scaffold vector. For toe-print assays, 5'-UTR-IRES with the

extended ORF part for primer annealing was cloned by a similar strategy. The uACG-AGA mutant was obtained by site-directed mutagenesis of 5'-UTR-IRES. T7 RNA polymerase in vitro transcription and purification on Spin-50 mini-column (USA Scientific) were used to obtain highly purified RNAs.

## Purification of translation components and ribosomal subunits

Native 40S subunits, eIF2, eIF3, eIF5B and rabbit aminoacyl-tRNA synthetases were prepared as previously described (*Pestova and Hellen, 2005*). Recombinant eIF1 and eIF1A were purified according to a previously described protocol (*Kolupaeva et al., 2007*). In vitro transcribed Met-tRNA$_i^{Met}$ was aminoacylated with methionine in the presence of rabbit aminoacyl-tRNA synthetases as previously described (*Pisarev et al., 2010*).

## Assembly of ribosomal complexes

To reconstitute different ribosomal complexes for toe-print assays, we incubated 0.3 pmol 5'-UTR-IRES RNA with 1.8 pmol 40S subunits, 10 pmol eIF1, 10 pmol eIF1A, 10 pmol eIF2, 5 pmol eIF3, 5 pmol eIF5B, and 5 pmol Met-tRNA$_i^{Met}$, as indicated, in a 20 µL reaction mixture containing buffer A (20 mM Tris-HCl [pH 7.5], 100 mM KCl, 2.5 mM MgCl$_2$ and 1 mM DTT) with 0.4 mM GTP and 0.4 mM ATP for 10 min at 37∘C. We analyzed the assembled ribosomal complexes via a toe-print assay, essentially as described by *Pestova and Hellen (2005)*. For the sucrose density gradient experiment, we incubated [$^{32}$P]-labelled 5'-UTR-IRES or HCV-IRES RNAs co-transcriptionally with 3.7 pmol 40S subunits and 11 pmol eIF3, as indicated, in a 60 µL reaction mixture containing buffer A for 10 min at 37°C, subjected the samples to a 10–30% sucrose density gradient centrifugation, and analyzed the gradient fractions by radioactivity counting.

## Cryo-EM sample preparation and data acquisition

Aliquots of 3 µl of assembled ribosome complexes at a concentration range of 250–350 nM were incubated for 30 s on glow-discharged holey gold grids (*Russo and Passmore, 2014*) (UltrAuFoil R1.2/1.3). Grids were blotted for 2.5 s and flash cooled in liquid ethane using a FEI Vitrobot. Grids were transferred to a FEI Titan Krios microscope equipped with an energy filter (slits aperture 20 eV) and a Gatan K2 detector operated at 300 kV. Data were recorded in counting mode at a magnification of 130,000, corresponding to a calibrated pixel size of 1.08 Å. Defocus values ranged from 1 µm to 3.6 µm. Images were recorded in automatic mode using the Leginon (*Carragher et al., 2000*) and APPION (*Lander et al., 2009*) software and frames were aligned using the Relion3 (*Zivanov et al., 2018*) implementation of the Motioncor2 algorithm (*Zheng et al., 2017*).

## Image processing and structure determination

Contrast transfer function parameters were estimated using GCTF (*Zhang, 2016*), and particle picking was performed using GAUTOMACH without the use of templates and with a diameter value of 260 pixels. All 2D and 3D classifications and refinements were performed using RELION. An initial 2D classification with a 4 times binned dataset identified all ribosome particles. A consensus reconstruction with all 40S particles was computed using the AutoRefine tool of RELION. Next, 3D classification without alignment (four classes, T parameter 4) identified a class with unambiguous density for eIF3. This class was independently refined, and further masked classification allowed the identification of two subclasses that are distinguishable by a different degree of 40S head swiveling and by the presence or absence of eIF3d density. Final refinements with unbinned data for the selected classes yielded high-resolution maps with density features in agreement with the reported resolution. Local resolution was computed with RESMAP (*Kucukelbir et al., 2014*).

## Model building and refinement

Models for the mammalian 40S and eIF3 docked into the maps using CHIMERA (*Pettersen et al., 2004*) and COOT (*Emsley et al., 2010*) were used to adjust these initial models manually. 5'-UTR-IRES was built manually using COOT. An initial round of refinement was performed in Phenix using real-space refinement (*Afonine et al., 2018*) with secondary structure restraints and a final step of reciprocal-space refinement with REFMAC (*Murshudov et al., 1997*). The fit of the model to the

map density was quantified using FSCaverage and Cref and model-to-maps over-fitting tests were performed following standard protocols in the field (*Brown et al., 2015*; *Amunts et al., 2014*).

**Cryo-EM data collection, refinement and validation statistics**

| | Class-1 (open) (EMDB-21529) (PDB 6W2S) | Class-2 (closed) (EMDB-21530) (PDB 6W2T) |
|---|---|---|
| **Data collection and processing** | | |
| Magnification | 130,000 | |
| Voltage (kV) | 300 | |
| Electron exposure (e–/Å²) | 59.55 | |
| Defocus range (μm) | −1 /– 3 | |
| Pixel size (Å) | 1.06 | |
| Symmetry imposed | C1 | |
| Initial particle images (no.) | 915,647 | |
| Final particle images (no.) | 14,257 | 23,444 |
| Map resolution (Å) | 3.3 | 3.3 |
| FSC threshold | 0.143 | 0.143 |
| Map resolution range (Å) | 3–8 | 3–8 |
| **Refinement** | | |
| Initial model used (PDB code) | 5A2Q | 5A2Q |
| Model resolution (Å) | 3.6 | 3.6 |
| FSC threshold | 0.5 | 0.5 |
| Model resolution range (Å) | 3.3–8 | 3.3–8 |
| Map sharpening $B$ factor (Å²) | −31.94 | −43.41 |
| Model composition | 106,817 | 109,684 |
| Non-hydrogen atoms | - | - |
| Ligands | | |
| $B$ factors (Å²) | 92.47 | 96.5 |
| Protein | 114.1 | 117.4 |
| RNA | | |
| R.m.s. deviations | 0.014 | 0.014 |
| Bond lengths (Å) | 1.77 | 1.78 |
| Bond angles (°) | | |
| Validation | 2.12 | 1.99 |
| MolProbity score | 6.13 | 4.92 |
| Clashscore | 1.62 | 1.39 |
| Poor rotamers (%) | | |
| Ramachandran plot | 88.92 | 90.25 |
| Favored (%) | 98.37 | 98.50 |
| Allowed (%) | 1.63 | 1.50 |
| Disallowed (%) | 0.18 | 0.17 |
| RNA validation | 2.35 | 2.25 |
| Angles outliers (%) | 0.442 | 0.428 |
| Sugar puckers outliers (%) | | |
| Average suit | | |

## Acknowledgements

We are grateful to Dr Jean-Luc Imler for a generous donation of a CrPV-5′-UTR-IRES plasmid. We are thankful to Prof. Jennifer Doudna for a generous donation of an HCV-IRES transcription vector. We are thankful to Prof. Kathrin Lang for the identification of an error in *Figure 4* in the pre-print version of this manuscript. We are thankful to Bob Grassucci and Zhening Zhang for assistance in cryo-EM data acquisition. This work was supported by the NIH National Institute of General Medical Sciences (GM097014 to AVP). Part of this work was performed at the Simons Electron Microscopy Center and National Resource for Automated Molecular Microscopy located at the New York Structural Biology Center, supported by grants from the Simons Foundation (SF349247), NYSTAR, and the NIH National Institute of General Medical Sciences (GM103310).

## Additional information

### Funding

| Funder | Grant reference number | Author |
|---|---|---|
| Columbia University | Start package | Israel S Fernández |
| National Institute of General Medical Sciences | GM097014 | Andrey V Pisarev |

The funders had no role in study design, data collection and interpretation, or the decision to submit the work for publication.

### Author contributions

Ritam Neupane, Software, Validation, Investigation, Visualization, Writing - review and editing; Vera P Pisareva, Investigation, Writing - review and editing; Carlos F Rodriguez, Investigation, carried out experiments; Andrey V Pisarev, Conceptualization, Investigation, Writing - original draft, Writing - review and editing; Israel S Fernández, Conceptualization, Resources, Supervision, Funding acquisition, Validation, Investigation, Visualization, Methodology, Project administration

### Author ORCIDs

Ritam Neupane (iD) https://orcid.org/0000-0003-4787-279X
Carlos F Rodriguez (iD) http://orcid.org/0000-0001-9166-0132
Israel S Fernández (iD) https://orcid.org/0000-0001-7218-1603

### Decision letter and Author response

Decision letter https://doi.org/10.7554/eLife.54575.sa1
Author response https://doi.org/10.7554/eLife.54575.sa2

## Additional files

### Supplementary files

• Transparent reporting form

### Data availability

Atomic coordinates have been deposited in the PDB with accession numbers and 6W2S and 6W2T for the open and closed classes , respectively . CryoEM maps have been deposited at the EMDB with accession numbers EMDB 21529 and 21530 for the open and closed classes respectively.

The following datasets were generated:

| Author(s) | Year | Dataset title | Dataset URL | Database and Identifier |
|---|---|---|---|---|
| Neupane R, Pisareva VP, Rodriguez CF, Pisarev AV, Fernández IS | 2020 | CryoEM map open class | https://www.ebi.ac.uk/pdbe/entry/emdb/EMD-21529 | Electron Microscopy Data Bank, EMD-21529 |
| Neupane R, Pisareva VP, Rodriguez CF, Pisarev AV, Fernández IS | 2020 | CryoEM map closed class | https://www.ebi.ac.uk/pdbe/entry/emdb/EMD-21530 | Electron Microscopy Data Bank, EMD-21530 |
| Neupane R, Pisareva VP, Rodriguez CF, Pisarev AV, Fernández IS | 2020 | Structure of the Cricket Paralysis Virus 5-UTR IRES (CrPV 5-UTR-IRES) bound to the small ribosomal subunit in the open state (Class 1) | https://www.rcsb.org/structure/6W2S | RCSB Protein Data Bank, 6W2S |
| Neupane R, Pisareva V, Rodriguez CF, Pisarev A, Fernandez IS | 2020 | Structure of the Cricket Paralysis Virus 5-UTR IRES (CrPV 5-UTR-IRES) bound to the small ribosomal subunit in the closed state (Class 2) | https://www.rcsb.org/structure/6W2T | RCSB Protein Data Bank, 6W2T |

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
