## [Decision Letter]

**Acceptance summary:**

We anticipate that your novel and exciting structures of a viral IRES will be of broad interest because they show how this virus binds initiation factors and wraps around the head of the 40S ribosome to position AUG codons in or near the P-site. Your structures resolved three domains of the IRES making specific contacts with the eIF3 complex and the 40S ribosome; of theses, domain II of the IRES, in particular, engages eIF3 and the 40S subunit RACK1 – explaining the atomic basis of this virus' dependence on RACK1.

**Decision letter after peer review:**

Thank you for submitting your article "A complex IRES at the 5'-UTR of a viral mRNA assembles a functional 48S complex via an uAUG intermediate" for consideration by *eLife*. Your article has been reviewed by three peer reviewers, one of whom is a member of our Board of Reviewing Editors, and the evaluation has been overseen by John Kuriyan as the Senior Editor. The following individual involved in review of your submission has agreed to reveal their identity: and Andrei A Korostelev (Reviewer #2).

The reviewers have discussed the reviews with one another and the Reviewing Editor has drafted this decision as an integration of everyone's feedback to help you prepare a thoroughly revised submission.

Summary:

Neupane and Pisareva et al. report novel and exciting structural insights into vRNA IRES structures from the cricket paralysis virus in different stages of hijacking mammalian ribosomes and initiation factors – and doing so in unexpected ways given prior publications (Gross et al., 2017). These authors utilize a biochemical reconstitution system based on rabbit reticulocyte lysates and purified components with multi-class cryo-EM to determine structures that suggest the CrPV 5'-UTR-IRES is more intricate than the simple type III classification proposed earlier. Specifically, these authors determined two IRES-engaged 48S complex structures in both an "open" state (Class^-1^) and second "closed" state (Class-2). Three domains of the IRES (Domain I, II and III) were resolved making specific contacts with the 40S and with the multi-subunit factor eIF3. Of particular interest, the authors reveal domain II of the IRES engages eIF3 as well as critical contacts between the IRES and RACK1-explaining the atomic basis of this IRES's dependence on RACK1. They also discuss how this IRES from CrPV requires eIF1/eIF1A and ternary complex-mediated Met-tRNAi delivery to transition from an upstream AUG (uAUG) to a translation-ready assembly on the start codon-including a plausible reaction coordinate for 40S conformational remodeling, eIF3(d) recruitment, and TC-Met-tRNAi delivery.

The use of mammalian components to study a cricket virus, however, raises some questions and concerns. 5'UTR IRESes are not conserved but rather highly divergent and can be quite specific for each virus. This IRES in the 5' UTR may have play a role in defining, at least partially, host tropism. Ideally, an IRES should be studied in a biologically relevant context using the translational machinery and ribosomes from a natural host – this is not the case with rabbit extracts and this discrepancy should be addressed prior to publication because it could explain a potential disagreement between the structural models and the toe-printing data.

Essential revisions:

1) The complex was found to be assembled on an out-of-frame upstream AUG (in the P-site of the 40S, so the authors propose a novel IRES-mediated mechanism using an uAUG as an intermediate complex that is required to further assemble the ribosome on the genuine AUG start codon. In order to strengthen their model, the authors performed toe-printing assays that allowed the detection of a toe-print on the uAUG and another toe-print on the genuine AUG. However, the confusing presentation of this data suggests there may be a contradiction with their model that warrants further exploration. Specifically, in Figure 6A it is unclear if the toeprint corresponding to uAUG (lanes 2-5) reflects tRNA bound to uAUG? It is possible that this band corresponds to the 40S-IRES-eIF3 complex (featured in cryo-EM structures), while tRNA loading does not occur despite the presence of the ternary complex? A control lane (with 40S, IRES and eIF3, without TC), should help to test this point and clarify the mechanism. In the current figure, there appear to be two toe-prints at +17/18 from the uAUG and the genuine AUG, respectively. Are these the expected distances if the uAUG is located in the P-site of the ribosome? If the uAUG is located in the A-site, shouldn't the corresponding toe-print be detected at +13/14? Please clarify. Moreover, the appearance of the genuine AUG toe-print is not correlated with the disappearance of the uAUG toe-print, which seems to indicate that the two toe-prints are independent. The authors propose that the uAUG toe-print corresponds to an intermediate state prior genuine AUG recognition. According to the toe-printing data, this does not seem to be the case and suggest that the structures may not correspond with the functional data.

2) To our knowledge, there is are no evidence in the literature that the uAUG is indeed required for this IRES to function, so the authors should mutate the uAUG and measure whether (i) the IRES is no longer active and (ii) whether both toe-prints disappear.

3) A biochemical test for the proposed roles of the DIII non-canonical base pairs would also strengthen the manuscript. Do mutants in this region of the IRES alter the positioning of the single-stranded segment of the IRES harboring the uAUG-701, and thus alter the outcomes of the toe-printing assay?

4) These authors show the 5'UTR IRES interacting directly with eIF3. Since there are significant differences between rabbit eIF3 and eIF3 from insects, this could explain why eIF3 is positioned differently on the 40S when compared canonical 48S structures. Please address.

5) The FSC 0.5 cut-off for the map-to-model correspondence appears to be notably worse than the half map 0.143 cut-off. The curve also has a very sharp drop-off toward zero, suggesting that the analysis was done with a low-pass filtered map or model? Also, please add a color legend for the refined B-factors.

6) The structural models have high clash scores, many poor rotamers, and far too many disallowed Ramachandran angles. Is this evidence of overfitting to density or other forms inattention to model geometry? Are there regions of the model that should not have been modeled at this level of detail?

7) The authors wrote "Further assistance by eIF1 and eIF1A…". But are there data suggesting such a step-wise IRES-dependent sequence of events? What data suggest that binding of eEF1 and eEF1A is excluded prior to binding of eIF3, IRES and/or delivery of tRNA? Perhaps eIF1 and/or eIF1A can be present prior to IRES and tRNA binding? Please discuss.

[Editors' note: further revisions were suggested prior to acceptance, as described below.]

Thank you for submitting your article "A complex IRES at the 5'-UTR of a viral mRNA assembles a functional 48S complex via an uAUG intermediate" for consideration by *eLife*. Your article has been reviewed by three peer reviewers, including Adam Frost as the Reviewing Editor and Reviewer #1, and the evaluation has been overseen by John Kuriyan as the Senior Editor. The following individual involved in review of your submission has agreed to reveal their identity: Andrei A Korostelev (Reviewer #2).

The reviewers have discussed the reviews with one another and the Reviewing Editor has drafted this decision to help you prepare a revised submission.

We would like to draw your attention to changes in our revision policy that we have made in response to COVID-19 (https://elifesciences.org/articles/57162). Specifically, when editors judge that a submitted work as a whole belongs in *eLife* but that some conclusions require a modest amount of additional new data, as they do with your paper, we are asking that the manuscript be revised to either limit claims to those supported by data in hand, or to explicitly state that the relevant conclusions require additional supporting data. Our expectation is that the authors will eventually carry out the additional experiments and report on how they affect the relevant conclusions either in a preprint on bioRxiv or medRxiv, or if appropriate, as a Research Advance in *eLife*, either of which would be linked to the original paper.

Summary:

The novel and exciting structural insights into an IRES from the cricket paralysis virus reported in the revised study by Neupane and Pisareva et al. shows this viral RNA hijacking mammalian ribosomes and initiation factors-and doing so in unforeseen ways given prior publications (Gross et al., 2017). The authors utilized a biochemical reconstitution system based on rabbit reticulocyte lysates and purified components, with multi-class cryo-EM, to determine structures of an "open" and "closed" state of the 40S ribosome bound by the IRES and eIF3. Three domains of the IRES (Domain I, II and III) were resolved making specific contacts with the eIF3+40S assembly. Domain II of the IRES engages eIF3 and makes contacts between the IRES and RACK1-explaining the atomic basis of this IRES's dependence on RACK1.

Each reviewer was intrigued by these novel structures because they show the viral RNA wrapping around the head of the 40S ribosome to position AUG codons in or near the P-site. Each reviewer, however, also expressed concerns about the mechanistic interpretation of viral translation initiation suggested by these structures-especially the unclear role of the upstream start codon or uAUG. After extensive cross-review, the reviewers consolidated the following requests to address before acceptance.

Essential revisions:

1) Interpretation of the uAUG's role and the toe-printing data

1a) The use of rabbit ribosomes and initiation factors to study an insect virus raised questions and concerns during the first round of review that the authors only partially addressed. One primary concern was that the possibility that the upstream AUG, or uAUG, may not be required for IRES function. The authors did not include a translational control-like reporter expression downstream of the IRES-which would enable functional evaluation of the role of the uAUG. This omission leaves open the possibility that the structures are of non-functional complexes that form in the absence of the ternary complex. The newly added controls show that mutation of the uAUG in the presence of the ternary complex, eIF2GTP+tRNAMeti, still leads to a robust toe-print on the bonafide AUG codon. All of the initiation factors under investigation here are present in cells-including eIF1, eIF1A, eIF3, the ternary complex, and eIF5B. The uAUG of the IRES, therefore, may not be not required for initiation in vivo even though in this system a stable toe-print forms in vitro. The authors should rewrite the description of their model, Discussion, sixth paragraph, to acknowledge these unresolved caveats and state clearly that the mechanism of initiation remains incompletely understood. Fully understanding initiation in this unusual setting (the staged roles of each initiation factor acting on the uAUG start-stop, followed by a genuine AUG start) will require further functional studies. Given the mismatch between host and virus, moreover (mammal versus insect), it is not clear that additional mechanistic experiments in the rabbit reticulocyte lysate context are worthwhile. To help explain this latter point, please include the percent similarity/identity between the rabbit and insect factors.

1b) The author's description and annotation of the location of uAUG and DIII in their structures (subsection “Non-canonical base pairing in the 5'-UTR-IRES DIII places the uAUG codon near the P site”) remain confusing. Please label the putative position of the uAUG in the middle panel of Figure 4C, along with the numbering for DIII, so it is clear to the reader where residue 701 could be. Since the uAUG bases themselves are not resolved in the map, the authors could highlight and label the final residue resolved in the P site or use a dotted line to schematically represented unresolved IRES bases to denote their putative positions.

1c) While the authors show that this IRES does not associate with the 40S without eIF3, the Figure 6 model figure shows "40S capture" rather than 40S+eIF3 capture-please fix.

1d) Also, in Figure 6, the 40S orientation changes too drastically between steps. It appears that the IRES and eIF3 jump between binding sites. Please fix the view of the 40S subunit view to help visualize conformational differences between the proposed initiation steps?

2) Quality and reliability of the atomic models: Concerns about overfitting remain due to the map-to-model FSC curves, the number of poor rotamers, and disallowed Ramachandran angles. The authors make salient points about the challenges of model building in this resolution regime and with this degree of heterogeneity, but they could be more conservative. Since their cryoEM density is not at sufficient resolution to justify clashes or bad geometry, we recommend "stumping" the residues/bases in question to the backbone to remove unsupported model features. In addition, the authors built their models using uniformly sharpened maps, despite the heterogeneous resolution. The models may improve, and their correspondence with the EM density may also improve, if the maps are first filtered and sharpened according to local resolution estimates. Locally filtered and sharpened maps may be better for modeling both high-resolution and low-resolution regions. Also, map-to-model FSC tests can certainly employ masks, but not low-pass filtered maps or models. Relion's default mode postprocessed maps are sharpened and low-pass filtered according to the global FSC. Please repeat map-model correspondence tests with unsharpened, unfiltered maps. There are many ways of generating such plots. Phenix.real_space_refine tools, for example, employ the unfiltered and unsharpened half-maps during model refinement and map-to-model correspondence tests. Finally, the table should include protein and RNA statistics-RNA statistics are currently missing.

---

## [Author Response]

Summary:Neupane and Pisareva et al. report novel and exciting structural insights into vRNA IRES structures from the cricket paralysis virus in different stages of hijacking mammalian ribosomes and initiation factors – and doing so in unexpected ways given prior publications (Gross et al., 2017). These authors utilize a biochemical reconstitution system based on rabbit reticulocyte lysates and purified components with multi-class cryo-EM to determine structures that suggest the CrPV 5'-UTR-IRES is more intricate than the simple type III classification proposed earlier. Specifically, these authors determined two IRES-engaged 48S complex structures in both an "open" state (Class^-1^) and second "closed" state (Class-2). Three domains of the IRES (Domain I, II and III) were resolved making specific contacts with the 40S and with the multi-subunit factor eIF3. Of particular interest, the authors reveal domain II of the IRES engages eIF3 as well as critical contacts between the IRES and RACK1-explaining the atomic basis of this IRES's dependence on RACK1. They also discuss how this IRES from CrPV requires eIF1/eIF1A and ternary complex-mediated Met-tRNAi delivery to transition from an upstream AUG (uAUG) to a translation-ready assembly on the start codon-including a plausible reaction coordinate for 40S conformational remodeling, eIF3(d) recruitment, and TC-Met-tRNAi delivery.The use of mammalian components to study a cricket virus, however, raises some questions and concerns. 5'UTR IRESes are not conserved but rather highly divergent and can be quite specific for each virus. This IRES in the 5' UTR may have play a role in defining, at least partially, host tropism. Ideally, an IRES should be studied in a biologically relevant context using the translational machinery and ribosomes from a natural host – this is not the case with rabbit extracts and this discrepancy should be addressed prior to publication because it could explain a potential disagreement between the structural models and the toe-printing data.

We certainly understand this concern and from a purely biologically point of view, yes, we agree the best scenario would be to study an IRES in the context of ribosomes from their own host. However, one of the aspects that makes the biology of IRESs so interesting is their ability to operate in a wide range of hosts and indeed such capability has been exploited for biotechnological applications. Two prototypical, widely studied IRESs, the IRES of the intergenic region of *Dicistroviruses* as well as the EMCV IRES, can manipulate ribosomes from different hosts and this capability has been exploited for gene expression regulation in multiple applications. With that in mind, we decided to explore the possibility the 5'-UTR-IRES of the CrPV would indeed function with mammalian components. Such strategy offers two advantages: on one side we will prove this IRES does not exclusively work in insects and second, and perhaps more importantly, we benefit from the solid knowledge both structural and biochemical, available for the mammalian system. There are dozens of high-resolution structures of mammalian ribosomes in several, biologically relevant configuration as well as a wealth of biochemistry and molecular biology data. All this knowledge is not available for the insect system.

Additionally, the ribosomal and eIF3 components the 5'-UTR-IRES interact with are highly conserved. Ribosomal protein uS17 or the core subunits of eIF3 are almost identical between insects and mammals, what argues against a "species" specific IRES. The final argument to prove the validity of our approach is the fact that toe-print experiments with mammalian components showed a very robust and selective delivery of initiator tRNA to 40S/5'-UTR-IRES/eIF3 complexes. This very specific and precise experiment will never work if the 5'-UTR-IRES do not manipulate mammalian 40S and eIF3 in a very specific way.

Essential revisions:1) The complex was found to be assembled on an out-of-frame upstream AUG (in the P-site of the 40S, so the authors propose a novel IRES-mediated mechanism using an uAUG as an intermediate complex that is required to further assemble the ribosome on the genuine AUG start codon. In order to strengthen their model, the authors performed toe-printing assays that allowed the detection of a toe-print on the uAUG and another toe-print on the genuine AUG. However, the confusing presentation of this data suggests there may be a contradiction with their model that warrants further exploration. Specifically, in Figure 6A it is unclear if the toeprint corresponding to uAUG (lanes 2-5) reflects tRNA bound to uAUG? It is possible that this band corresponds to the 40S-IRES-eIF3 complex (featured in cryo-EM structures), while tRNA loading does not occur despite the presence of the ternary complex? A control lane (with 40S, IRES and eIF3, without TC), should help to test this point and clarify the mechanism. In the current figure, there appear to be two toe-prints at +17/18 from the uAUG and the genuine AUG, respectively. Are these the expected distances if the uAUG is located in the P-site of the ribosome? If the uAUG is located in the A-site, shouldn't the corresponding toe-print be detected at +13/14? Please clarify. Moreover, the appearance of the genuine AUG toe-print is not correlated with the disappearance of the uAUG toe-print, which seems to indicate that the two toe-prints are independent. The authors propose that the uAUG toe-print corresponds to an intermediate state prior genuine AUG recognition. According to the toe-printing data, this does not seem to be the case and suggest that the structures may not correspond with the functional data.

We appreciate the reviewer's concern about the explanation of these results. Toe-printing experiments are difficult to understand, and a better explanation is needed here. We completely rewrote the section where these experiments are described, citing Skabkin et al.paper. We hope the experiments, which show robust, unambiguous results, could be interpreted better now, especially by the reader not familiarized with this kind of data.

Toe-print signals can only be identified if the ribosome is solidly anchored to the mRNA. This will stop the reverse polymerase. No toe-print signal around any of the AUG codons could be observed for the 5'-UTR-IRES alone, 40S/5'-UTR-IRES complex or the 40S/5'-UTR-IRES/eIF3 complex which was characterized structurally by cryoEM. These controls were very clear and are now included in a supplementary figure (Figure 6—figure supplement 1). Second, there are two clear toe-print signals arising from the delivery of initiator Met-tRNA_i_^Met^ to the P site of the ribosome. Both eIF2 and eIF5B can only deliver Met-tRNA_i_^Met^ to the P site of the 40S, where they establish specific interaction and are able to stabilize the Met-tRNA_i_^Met^ in a solid conformation able to generate a toe-print. If this tRNA/mRNA/ribosome interaction is not solid, the reverse polymerase is able to disassemble the weak complex, keeping elongating and no toe-print signal could be observed. That is the case for Figure 6A lane 1: Met-tRNA_i_^Met^ by itself is unable to find the right ribosomal context so it can't establish interactions with the mRNA/40S stable enough to generate a toe-print around the AUGs. It is well known that P site toe-prints generate signals 15-17 nucleotides downstream due to that is the length of mRNA covered by the 40S. In our experiments, we see two toe-prints exactly 15/17 nucleotides away from the uAUG or the annotated AUG, what makes the interpretation of the results unambiguous. No Met-tRNA_i_^Met^ has ever been described to be able to generate a toe-print at 13/14 due to it is pausing the ribosome in the A site.

Band intensity in toe-print experiments are not quantitative, they do not reflect a total amount of radioactivity present in the assay, specially, when we compare experiments with different components like lanes 2-5 in Figure 6A. The main point we want to clarify with these toe-print experiments is how the 5'-UTR-IRES is able to direct by itself Met-tRNA_i_^Met^ to the P site of the 40S. This delivery event is very specific and only takes place in very determined cellular conditions, normally associated with covalently modified messengers ("capped mRNAs") and the initiator factors of the eIF4 family, absent here. The fact that 5'-UTR-IRES, in conjunction only with eIF2 or eIF5B can generate, solid, robust toe-print signals on AUGs is remarkable, and can only happen due to the ability of this RNA sequence to specifically manipulate 40S and initiation factors. Importantly, this IRES operate in a cellular environment where general translation has been shut off due to phosphorylation of eIF2alpha. In this context, eIF5B can substitute eIF2 in Met-tRNA_i_^Met^ delivery, however, how can that happen is not fully understood. In our experiments, we can only see delivery of Met-tRNA_i_^Met^ by eIF5B to the uAUG (Figure 6A lane 4). This delivery is less efficient when eIF1/eIF1A are present (Figure 6A lane 5). eIF2 also delivers Met-tRNA_i_^Met^ to the uAUG in isolation but, in contrast to eIF5B, in the presence of eIF1/eIF1A it can find both, the uAUG and the annotated AUG, both with more efficiency as judged by the toe-print intensity than the recognition of uAUG in isolation. This represent only part of the truth as all these factors act in vivo in conjunction and with others. We believe these results indicate a dynamic interplay between different initiation factors able to deliver the initiator tRNA to different but physically close AUG codons, and such dynamisms could be essential for an efficient initiation in a cellular context where cellular and viral messages compete for ribosomal access.

We do feel that the fact that two different initiation factors can deliver Met-tRNA_i_^Met^ to this mRNA is remarkable but clearly, more experiments are needed to fully understand this mechanism.

2) To our knowledge, there is are no evidence in the literature that the uAUG is indeed required for this IRES to function, so the authors should mutate the uAUG and measure whether (i) the IRES is no longer active and (ii) whether both toe-prints disappear.

As we discussed above, there are several factors to have in mind regarding the importance of the uAUG and start-stop configuration that precedes the annotated AUG. We indeed mutate this codon to ACG (new supplementary Figure 6—figure supplement 1B) and, as expected, the delivery of Met-tRNA_i_^Met^ to uAUG by eIF2 and eIF1/eIF1A is abolished, while the delivery to the annotated AUG is maintained (Figure 6—figure supplement 1B, right lane). However, we are skeptical regarding the importance of this. As noted above, in a cellular environment where eIF2 is not available due to phosphorylation of its alpha subunit, only eIF5B has been reported to substitute eIF2 in Met-tRNA_i_^Met^ delivery. Thus, in a biologically relevant situation this mutation will probably abolish initiation mediated by the 5'-UTR-IRES.

3) A biochemical test for the proposed roles of the DIII non-canonical base pairs would also strengthen the manuscript. Do mutants in this region of the IRES alter the positioning of the single-stranded segment of the IRES harboring the uAUG-701, and thus alter the outcomes of the toe-printing assay?

Unfortunately, we are unable to generate these mutants and test them by toe-print in the time frame expected for the return of this reviewed version. We strongly believe this mutation will be deleterious for the activity of the IRES as they would radically change the local architecture of the IRES at a critical region.

4) These authors show the 5'UTR IRES interacting directly with eIF3. Since there are significant differences between rabbit eIF3 and eIF3 from insects, this could explain why eIF3 is positioned differently on the 40S when compared canonical 48S structures. Please address.

In our maps and derived models, eIF3 exhibit a positioning respect the 40S very similar to the position described for the canonical 48S complex (Figure 3D). The 5'-UTR-IRES seems to exploit cavities present in the 48S canonical complex to interact with the 40S-facing area of eIF3. In any case, there are two arguments that convince us the interaction of the 5'-UTR-IRES with the mammalian versus insect eIF3 should be similar: on one side, the interaction of DII of 5'-UTR-IRES with the back of the 40S and eIF3 is peripherical, with very limited contacts between DII and eIF3, mediated by contacts with the phosphate backbone of the IRES and not with specific bases. On the other hand, and perhaps more importantly, eIF3 core subunits a and c (that form the area hosting the bulk of interactions with 5'-UTR-IRES DII) are highly conserved, as it is ribosomal protein uS17 at the back of the 40S.

5) The FSC 0.5 cut-off for the map-to-model correspondence appears to be notably worse than the half map 0.143 cut-off. The curve also has a very sharp drop-off toward zero, suggesting that the analysis was done with a low-pass filtered map or model? Also, please add a color legend for the refined B-factors.

We understand the concern regarding the discrepancy between the model-vs.-map FSC 0.5 cut-off and half-map 0.143 cut-off as they are indeed not as close as they should be for this resolution. The explanation is related to the wide heterogeneity of the map, even after classification. The map exhibits very good quality at the 40S body and some areas of the 40S head, some areas of the IRES and the regions of the eIF3 in contact with the 40S. Poor density could however be observed in the regions of eIF3 away from the 40S and some segments of the IRES exposed to the solvent. In such scenario, current protocols of model refinement in cryoEM maps are far from ideal. We monitored the weight applied to the geometry term in the final step of the model refinement (using REFMAC) in order to apply as less as possible provided the model was not overfitted. It turned out that due to the areas of the map with severe flexibility, we had to keep the geometry term quite low in order to avoid overfitting, in our opinion a more severe problem that a discrepancy between the model-vs.-map FSC 0.5 cut-off and half-map 0.143 cut-off.

In the model validation FSC (Figure 1—figure supplement 2B and F, black curve) it was used the final, post-processed masked, map which was the same one used for model refinement. The masking artifact explain the drastic fall to zero. Due to this is the final map used for model building and refinement, we feel this is the one that should be used for the FSC calculation.

We appreciate the suggestion of adding labels for the model B-factor scale. These labels have been added.

6) The structural models have high clash scores, many poor rotamers, and far too many disallowed Ramachandran angles. Is this evidence of overfitting to density or other forms inattention to model geometry? Are there regions of the model that should not have been modeled at this level of detail?

As described in the previous point, the heterogeneity of the map imposed strict limitations on the model refinement strategies. We opted for a conservative approach, considering the avoiding of overfitting of the model to the map the most important objective. We performed 10 different model refinement runs for the two maps, adjusting the geometry-versus-map-fit (matrix keyword in REFMAC) and subsequently calculating a model "shake" and half map 1 refinement overfitting test. The final matrix value chose was the highest one that exhibited no overfitting. As shown by the overlapping of the red and blue curves of Figure 1—figure supplement 2B and F, both models are not overfitted.

Regarding the Ramachandran outliers, we consider these values reasonable given the resolution and when compared with other ribosome models at similar resolution both by cryoEM or X-ray crystallography. REFMAC does not implement Ramachandran restrains in its reciprocal-space refinement protocol, due to that could lead to artificially low Ramachandran outliers percentage. We believe this approach is best suited for the current state of the field instead of trying to obtain a 0% Ramachandran outliers score, unrealistic given this resolution and map heterogeneity.

7) The authors wrote "Further assistance by eIF1 and eIF1A…". But are there data suggesting such a step-wise IRES-dependent sequence of events? What data suggest that binding of eEF1 and eEF1A is excluded prior to binding of eIF3, IRES and/or delivery of tRNA? Perhaps eIF1 and/or eIF1A can be present prior to IRES and tRNA binding? Please discuss.

We have indeed no evidence for the sequential recruitment of eIF1/eIF1A after TC recruitment. We have modified our proposed model to reflect this fact, including Figure 6.

[Editors' note: further revisions were suggested prior to acceptance, as described below.]

Essential revisions:1) Interpretation of the uAUG's role and the toe-printing data1a) The use of rabbit ribosomes and initiation factors to study an insect virus raised questions and concerns during the first round of review that the authors only partially addressed. One primary concern was that the possibility that the upstream AUG, or uAUG, may not be required for IRES function. The authors did not include a translational control-like reporter expression downstream of the IRES-which would enable functional evaluation of the role of the uAUG. This omission leaves open the possibility that the structures are of non-functional complexes that form in the absence of the ternary complex.

We respectfully disagree with this concern. Every IRES need to first and foremost interact with the ribosomal subunits before any aminoacyl-tRNA is delivered. One of the most essential features of any IRES is its ability to capture ribosomal subunits in isolation or in combination with initiation factors. The best characterized IRESs (the IGR-IRES of the CrPV and HCV IRES) are both able to interact with the 40S and/or 80S in the absence of the TC. This is so because it is compulsory for these RNAs to first, bind and hijack the ribosomal subunits and only afterwards, recruit a tRNA delivery system either from initiation (eIF2/eIF5B) or from elongation (eEF2/eEF2A). Thus, the sentence:

“This omission leaves open the possibility that the structures are of non-functional complexes that form in the absence of the ternary complex.” in our opinion, is highly misleading, especially for the non-specialized reader, as it suggests that the interaction of the IRES with the 40S in the absence of the ternary complex could be interpreted as an “artifact” when in fact is an essential, biologically relevant step in IRES biology.

The newly added controls show that mutation of the uAUG in the presence of the ternary complex, eIF2GTP+tRNAMeti, still leads to a robust toe-print on the bonafide AUG codon. All of the initiation factors under investigation here are present in cells-including eIF1, eIF1A, eIF3, the ternary complex, and eIF5B. The uAUG of the IRES, therefore, may not be not required for initiation in vivo even though in this system a stable toe-print forms in vitro.

We consider this statement as an oversimplification given two facts: first, we disagree with the conclusion that due to the mutation of uAUG does not abolish the delivery of Met-tRNA_i_^Met^ to the bona-fide AUG codon by eIF2, uAUG is not required for initiation in vivo. In physiological conditions, if a cell is infected by a virus, it is well established that the availability of eIF2 is highly diminished. Phosphorylation of eIF2-alpha, caused by the antiviral response of the cell, blocks the delivery of Met-tRNA_i_^Met^ by eIF2. Then, how viral proteins initiate under these conditions? The most probable candidate to substitute eIF2 in Met-tRNA_i_^Met^ delivery is eIF5B and eIF5B, as our toe-prints clearly show, ONLY delivers Met-tRNA_i_^Met^ to the uAUG. Secondly, the uAUG is part of a start-stop configuration which means that perhaps re-initiation events mediated by release factors could be critical for sustained levels of translation initiation in infected cell. We agree with reviewers that more experiments, probably in vivo, are required to understand how this peculiar initiation happens, but we believe that ruling out a significant role for the uAUG, just because its mutation does not abolish the recognition of the bone-fide AUG by eIF2, seems premature.

The authors should rewrite the description of their model, Discussion, sixth paragraph, to acknowledge these unresolved caveats and state clearly that the mechanism of initiation remains incompletely understood. Fully understanding initiation in this unusual setting (the staged roles of each initiation factor acting on the uAUG start-stop, followed by a genuine AUG start) will require further functional studies.

We totally agree with this conclusion and we hope this study will inspire experiments directed towards the understanding of the role of uAUGs, eIF5B and the start-stop configuration in cellular and viral messengers. In order to reflect this, we have modified the text with the following sentence:

“However, these considerations are highly speculative, as the particular role the uAUG exerts in Met-tRNA_i_^Met^ recruitment or more generally, its involvement in initiation of viral messengers remains enigmatic. A comprehensive understanding of the role of uAUG and the start-stop configuration will demand further studies, ideally in vivo.”

Given the mismatch between host and virus, moreover (mammal versus insect), it is not clear that additional mechanistic experiments in the rabbit reticulocyte lysate context are worthwhile. To help explain this latter point, please include the percent similarity/identity between the rabbit and insect factors.

identity/similarity (%) (*Drosophila*/mammals)eIF2alpha:57/74beta:74/83gamma:82/88eIF5B:71/85eIF3a:46/63eIF3c:51/66

We have supplemented the manuscript with this comparative analysis.

1b) The author's description and annotation of the location of uAUG and DIII in their structures (subsection “Non-canonical base pairing in the 5'-UTR-IRES DIII places the uAUG codon near the P site”) remain confusing. Please label the putative position of the uAUG in the middle panel of Figure 4C, along with the numbering for DIII, so it is clear to the reader where residue 701 could be. Since the uAUG bases themselves are not resolved in the map, the authors could highlight and label the final residue resolved in the P site or use a dotted line to schematically represented unresolved IRES bases to denote their putative positions.

We followed the reviewer’s advice in modifying Figure 4C to include labels for the last residue of the IRES seen in density (C695) as well as, using as guide the mRNA trajectory from known structures, we now indicate with a dotted line the putative placement of the segment of viral mRNA harboring the uAUG.

1c) While the authors show that this IRES does not associate with the 40S without eIF3, the Figure 6 model figure shows "40S capture" rather than 40S+eIF3 capture-please fix.

We have eliminated this step from the figure and modified the orientation of the complexes to facilitate its viewing. We have also added labels for the IRES domains to allow a better perception of the transitions.

1d) Also, in Figure 6, the 40S orientation changes too drastically between steps. It appears that the IRES and eIF3 jump between binding sites. Please fix the view of the 40S subunit view to help visualize conformational differences between the proposed initiation steps?

These modifications have been introduced. See above.

2) Quality and reliability of the atomic models: Concerns about overfitting remain due to the map-to-model FSC curves, the number of poor rotamers, and disallowed Ramachandran angles. The authors make salient points about the challenges of model building in this resolution regime and with this degree of heterogeneity, but they could be more conservative. Since their cryoEM density is not at sufficient resolution to justify clashes or bad geometry, we recommend "stumping" the residues/bases in question to the backbone to remove unsupported model features. In addition, the authors built their models using uniformly sharpened maps, despite the heterogeneous resolution. The models may improve, and their correspondence with the EM density may also improve, if the maps are first filtered and sharpened according to local resolution estimates. Locally filtered and sharpened maps may be better for modeling both high-resolution and low-resolution regions.

We greatly value reviewers' input in this area. After, careful reading of previous concerns we have realized there were some mistakes in the values reported in the table summarizing cryoEM data. We have updated these values which are now correct, reflecting better statistics.

We started the model building and initial refinement in real space with locally refined maps using masks for the 40S-head-IRES-DI and 40S-head-IRES-DII as the reviewer suggests. These masked maps, as shown in Figure 1—figure supplement 2, locally improved the density allowing a more uniform post-processing. These local masked maps are now uploaded in the EMDB deposition.

However, we believe that the “real” final model should be refined against the final, whole map, especially in the reciprocal-space step when we use Refmac. This step allows for a realistic computation of B-factors in reciprocal space which, as shown in Figure 1—figure supplement 2, reflect accurately the variability of the maps seen by local resolution calculations. We believe this information is valuable for the specialized reader, who can quickly from these diagrams identify which regions of the maps/model are more trustable and which ones should be considered cautiously.

Our models are clearly not over-fitted. We performed a rigorous over-fitting test by refining against half-map-1 a “shake” model. The FSC curves between this re-refined model against half-map-1 and against half-map-2 (not included in the refinement) clearly overlap for both classes, what unambiguously demonstrates that our models are not over-fitted (over-fitting test described in: Amunts et al., 2014 and Brown, et al., 2015).

Also, map-to-model FSC tests can certainly employ masks, but not low-pass filtered maps or models. Relion's default mode postprocessed maps are sharpened and low-pass filtered according to the global FSC. Please repeat map-model correspondence tests with unsharpened, unfiltered maps. There are many ways of generating such plots. Phenix.real_space_refine tools, for example, employ the unfiltered and unsharpened half-maps during model refinement and map-to-model correspondence tests. Finally, the table should include protein and RNA statistics-RNA statistics are currently missing.

We have update the FSC for these graphs using non-filtered unsharpened maps in Figure 1—figure supplement 2.